# DETERMINISTIC DISCRETE DENOISING

## ABSTRACT

We propose a deterministic denoising algorithm for discrete-state diffusion models based on Markov chains. The generative reverse process is derandomized by introducing a variant of the herding algorithm with weakly chaotic dynamics, which induces deterministic discrete state transitions. Our approach is a direct replacement for the stochastic denoising process, requiring neither retraining nor continuous state embeddings. We demonstrate consistent improvements in both efficiency and sample quality on text and image generation tasks. Thus, this simple derandomization approach is expected to enhance the significance of discrete diffusion in generative modeling. Furthermore, our results reveal that deterministic reverse processes, well established in continuous diffusion, can also be effective in discrete state spaces.

## 1 INTRODUCTION

Diffusion models have recently achieved remarkable success in generating realistic image and audio data. Their forward process progressively corrupts data with noise, and the models are trained to denoise by inverting the corruption process. Once trained, diffusion models can generate realistic samples from pure noise by iteratively applying the denoising procedure. For continuous data such as images and speech, the reverse denoising process is often implemented as a deterministic algorithm, which exhibits more directed dynamics and enables efficient generation with fewer denoising steps compared to stochastic denoising, as demonstrated by DDIM (Song et al., 2021).

Diffusion models for discrete data are also an important research direction, since many data of interest are inherently discrete, including text, graphs, genomes, and chemical structures. In discrete-state diffusion models, the forward and reverse processes are typically formulated as probabilistic Markov chains on a discrete sample space (Austin et al., 2021). If the reverse denoising process could be made deterministic, improved performance would be expected, as in the case of continuous models. However, in discrete spaces, naive derandomization as a deterministic mapping aggregates discrete trajectories and reduces sample diversity, which is essential for generative modeling.

One promising approach is to embed the discrete sample space into a continuous space and apply continuous diffusion models (Hoogeboom et al., 2021; Chen et al., 2023; Sahoo et al., 2025). The main advantage of this approach is that it allows the direct use of well-established techniques for continuous diffusion, including deterministic denoising. Nevertheless, denoising in the continuous space requires retraining the model. Moreover, it remains unclear whether discrete data must be embedded into a continuous space for derandomization.

In this paper, we propose to directly derandomize the reverse denoising process on discrete sample spaces. The key idea is to introduce a variant of the herding algorithm (Welling, 2009; Welling & Chen, 2010), which associates auxiliary continuous weight variables with each discrete variable in the sample space. The reverse denoising updates of both discrete and continuous variables are deterministic, and randomness arises solely from their initial states. Since the proposed algorithm derandomizes the state transitions of Markov chains, it serves as a drop-in replacement for the stochastic reverse process without retraining discrete diffusion models. This relation is analogous to DDIM (Song et al., 2021), which generates samples deterministically using a model trained for DDPM (Ho et al., 2020). In our experiments on text and image generation, we demonstrate consistent improvements in both efficiency and sample quality. These results reveal that deterministic reverse processes, well established in continuous diffusion, can also be effective in discrete state spaces.

In the evaluation, we adopt UDLM (Schiff et al., 2025), a state-of-the-art discrete-state generative model based on the uniform diffusion process. Our aim in this paper is not to propose a new discrete diffusion model, but to demonstrate how the herding-based algorithm improves the generative process of existing ones. We use the official training, sampling, and evaluation codes of UDLM, replacing only the reverse denoising procedure. This directly shows that our method works as a drop-in replacement without retraining. Moreover, the uniform diffusion process is particularly suitable for our approach, since its reverse process performs iterative refinement through successive Markov state transitions, making it a natural setting to examine whether derandomizing these transitions can improve efficiency and sample quality.

In contrast, masked diffusion has been widely studied as a discrete diffusion model, but its reverse process allows each token to transition only once from the masked state and remain fixed thereafter, fundamentally limiting its dynamics. A related line of work is the deterministic reverse algorithm proposed in Chen et al. (2024), which performs deterministic state transitions in discrete sample spaces. This approach is effective for masked diffusion, where the transition time is limited. Our work complements this result by showing that deterministic denoising is also beneficial for uniform diffusion models, where state transitions continue throughout the reverse process.

Overall, we present a simple yet effective derandomization approach to directly enhance the generative process of existing discrete-state diffusion models, thus strengthening the significance of discrete diffusion in generative modeling.

## 2 BACKGROUND

### 2.1 DISCRETE DIFFUSION MODELS

A diffusion process in a discrete state space can be viewed as a sequence of discrete state transitions that gradually corrupt the original data. This process can be formulated as a discrete-time Markov chain over categorical variables, giving rise to a reverse-time denoising Markov chain (Austin et al., 2021). This formulation can be naturally extended to state transitions in continuous time, using Markov jump processes parameterized by transition-rate matrices (Campbell et al., 2022; Sun et al., 2023). Foundational theoretical studies analyze approximation error and convergence properties of both discrete-time and continuous-time discrete diffusion models (Ren et al., 2025a; Zhang et al., 2025; Liang et al., 2025a;b). Below, we focus on a discrete-time formulation to outline how our approach can be applied to the reverse denoising process.

We consider generative models for discrete data represented as a sequence of $L$ categorical tokens. Each token takes one of $K$ categories and is represented as a $K$-dimensional one-hot vector $\boldsymbol{x} \in \mathcal{V}$, where $\mathcal{V} = \{(x_1, x_2, \ldots, x_K)^\top \in \{0,1\}^K \mid \sum_{k=1}^K x_k = 1\}$. A sequence of $L$ tokens is denoted by $\boldsymbol{x}^{(1:L)} = (\boldsymbol{x}^{(1)}, \boldsymbol{x}^{(2)}, \ldots, \boldsymbol{x}^{(L)}) \in \mathcal{V}^L$.

The forward noising process in the D3PM framework (Austin et al., 2021) is formulated as a discrete-time Markov chain on $\mathcal{V}$, which gradually transforms the original data $\boldsymbol{x}_0$ into noise $\boldsymbol{x}_T$. Given the Markov transition matrix $Q_t$ at time $t$, the forward transition is defined as

$$q(\boldsymbol{x}_t \mid \boldsymbol{x}_{t-1}) = \mathrm{Cat}(\boldsymbol{x}_t; Q_t \boldsymbol{x}_{t-1}), \tag{1}$$

where $\mathrm{Cat}(\cdot; \boldsymbol{p})$ denotes the categorical distribution with the probability vector $\boldsymbol{p}$.

The Markov transition matrix $Q_t$ is designed to gradually transform any categorical distribution into a stationary distribution as $t$ increases. For the uniform noising process, it is given by

$$Q_t = (1 - \beta_t)\boldsymbol{I} + \beta_t \frac{1}{K} \mathbf{1}\mathbf{1}^\top, \tag{2}$$

where $\beta_t \in (0, 1)$ is a noise schedule. For the absorbing noising process, it is defined as

$$Q_t = (1 - \beta_t)\boldsymbol{I} + \beta_t \boldsymbol{x}_{\mathrm{mask}} \mathbf{1}^\top, \tag{3}$$

where $\boldsymbol{x}_{\mathrm{mask}}$ is the one-hot vector representing the absorbing category. In both cases, the Markov transition matrix from time $s$ to $t$, $Q_t Q_{t-1} \ldots Q_{s+1}$, admits a closed-form expression for arbitrary $s < t$, which can be extended to non-integer times. This property is exploited to improve the efficiency of training and sampling (Sahoo et al., 2024; Shi et al., 2024; Schiff et al., 2025).

The reverse denoising process is also formulated as a discrete-time Markov chain on $\mathcal{V}^L$. A neural network model parameterized by $\theta$ is trained to predict $\tilde{\boldsymbol{x}}_0^{(1:L)}$, an estimate of the original data $\boldsymbol{x}_0^{(1:L)}$, from noisy data $\boldsymbol{x}_t^{(1:L)}$ at time $t$. Training is performed by minimizing a variational upper bound (NELBO) on the negative log-likelihood (Austin et al., 2021; Schiff et al., 2025). The transition probability $p_\theta(\boldsymbol{x}_{t-1} \mid \boldsymbol{x}_t^{(1:L)}, t)$ of the reverse process is then computed from the predicted original data $\tilde{\boldsymbol{x}}_0^{(1:L)}$ and the forward transition matrix $Q_t$.

By iterating this Markov transition from pure noise $\boldsymbol{x}_T^{(1:L)}$ sampled from the stationary distribution, the reverse process generates a sample $\boldsymbol{x}_0^{(1:L)}$. Namely, each denoising step reduces to sampling from the categorical distribution $\mathrm{Cat}(\boldsymbol{x}_{t-1}; \boldsymbol{p}_{t-1})$ with probability vector $\boldsymbol{p}_{t-1} = p_\theta(\boldsymbol{x}_{t-1} \mid \boldsymbol{x}_t^{(1:L)}, t)$. This is typically implemented as a probabilistic procedure using the Gumbel-max trick. However, sampling does not necessarily have to be probabilistic; it suffices that samples are generated faithfully according to the given transition probabilities. This motivates us to explore deterministic alternatives for the reverse denoising process.

## 2.2 Herding Algorithm

The herding algorithm (Welling, 2009; Welling & Chen, 2010) is a deterministic sampling method with weakly chaotic dynamics, which combines statistical learning and inference for energy-based models into a single dynamical system.

The herding system yields a sample sequence $\boldsymbol{x}_1, \boldsymbol{x}_2, \ldots$ from a discrete sample space $\mathcal{V}$ such that the empirical average of feature values $\boldsymbol{\phi}(\boldsymbol{x}_t)$ converges to predefined target values $\boldsymbol{\mu} \in \mathbb{R}^N$, where $\boldsymbol{\phi} \colon \mathcal{V} \to \mathbb{R}^N$ is a set of real-valued feature functions.

To reduce the discrepancy between the averaged feature values and the target values in each sampling step, the herding system updates the weight vector $\boldsymbol{w}_t$. Specifically, the herding system generates a new sample $\boldsymbol{x}_{t+1}$ and updates the weight vector $\boldsymbol{w}_{t+1}$ at time $t+1$ as follows:

$$\boldsymbol{x}_{t+1} = \arg\max_{\boldsymbol{x} \in \mathcal{V}} \boldsymbol{w}_t^\top \boldsymbol{\phi}(\boldsymbol{x}), \tag{4}$$

$$\boldsymbol{w}_{t+1} = \boldsymbol{w}_t + \boldsymbol{\mu} - \boldsymbol{\phi}(\boldsymbol{x}_{t+1}). \tag{5}$$

This forms a hybrid dynamical system (Aihara & Suzuki, 2010) consisting of a discrete state variable $\boldsymbol{x}_t$ and a continuous state variable $\boldsymbol{w}_t$. Given an initial state $(\boldsymbol{x}_0, \boldsymbol{w}_0) \in \mathcal{V} \times \mathbb{R}^N$, the system generates samples iteratively, deterministically, and autonomously. Typically, the weight vector $\boldsymbol{w}_t$ does not converge and remains in a bounded domain around the origin.

The difference between the predefined target vector and the feature vector averaged over $T$ samples can be written as

$$\boldsymbol{\mu} - \frac{1}{T} \sum_{t=1}^{T} \boldsymbol{\phi}(\boldsymbol{x}_t) = \frac{1}{T}(\boldsymbol{w}_T - \boldsymbol{w}_0). \tag{6}$$

The right-hand side converges to zero as $T \to \infty$, provided that the sequence $\boldsymbol{w}_t$ remains bounded, which is guaranteed under mild conditions. Therefore, the herding system generates a sequence of samples with the predefined feature expectations converging asymptotically at a rate of $O(T^{-1})$. This convergence rate is much faster than the $O(T^{-1/2})$ rate of probabilistic sampling, such as independent categorical sampling and Markov chain Monte Carlo (MCMC) sampling, from a distribution with the same feature expectations. Note that exact maximization in the sample generation step is not required, since the $O(T^{-1})$ convergence holds as long as the weight vector remains bounded (Welling, 2009; Welling & Chen, 2010). The convergence rate of the herding algorithm is analyzed in more detail (Bach et al., 2012; Harvey & Samadi, 2014).

The dynamics of the herding system is classified as a piecewise isometry (Goetz, 2000; 2003), which is weakly chaotic, with Lyapunov exponents equal to zero almost everywhere, and typically has a fractal attracting set. Such complex nonlinear dynamics of the herding system yields diverse samples with relatively high entropy and negative autocorrelations (Welling, 2009; Welling & Chen, 2010).

For sampling from a categorical distribution $\mathrm{Cat}(\boldsymbol{x}; \boldsymbol{p})$ on $\mathcal{V}$, the feature function $\boldsymbol{\phi}(\boldsymbol{x}) = \boldsymbol{x}$ and the target vector $\boldsymbol{\mu} = \boldsymbol{p}$ lead to the following herding dynamics:

$$\boldsymbol{x}_{t+1} = \underset{\boldsymbol{x} \in \mathcal{V}}{\arg\max}\, \boldsymbol{w}_t^\top \boldsymbol{x}, \tag{7}$$

$$\boldsymbol{w}_{t+1} = \boldsymbol{w}_t + \boldsymbol{p} - \boldsymbol{x}_{t+1}. \tag{8}$$

This system generates samples from $\mathrm{Cat}(\boldsymbol{x}; \boldsymbol{p})$ whose empirical distribution is closer to $\boldsymbol{p}$ than that of probabilistic categorical sampling.

## 3 DERANDOMIZATION OF REVERSE DENOISING PROCESS

We apply the herding algorithm to the reverse denoising process of discrete diffusion models. For derandomization of the Markov chain, we need to introduce a time-dependent version of the herding algorithm as in (Suzuki, 2014).

Specifically, the sample $\boldsymbol{x}_t$ and weight vector $\boldsymbol{w}_t$ for each token are updated from time $t$ to $t-1$ as follows:

$$\boldsymbol{x}_{t-1} = \underset{\boldsymbol{x} \in \mathcal{V}}{\arg\max}\, (\boldsymbol{w}_t + \boldsymbol{p}_{t-1})^\top \boldsymbol{x}, \tag{9}$$

$$\boldsymbol{w}_{t-1} = \boldsymbol{w}_t + \boldsymbol{p}_{t-1} - \boldsymbol{x}_{t-1}, \tag{10}$$

where $\boldsymbol{p}_{t-1} = p_\theta(\boldsymbol{x}_{t-1} \mid \boldsymbol{x}_t^{(1:L)}, t)$ denotes the probability vector of the categorical distribution obtained from the model prediction. In Eq. (9), the current probability vector $\boldsymbol{p}_{t-1}$ is taken into account when selecting the new sample $\boldsymbol{x}_{t-1}$, by taking the argmax of $(\boldsymbol{w}_t + \boldsymbol{p}_{t-1})^\top \boldsymbol{x}$ instead of $\boldsymbol{w}_t^\top \boldsymbol{x}$. This herding-based system reduces the discrepancy $\sum_{\tau=t}^{T-1}(\boldsymbol{p}_\tau - \boldsymbol{x}_\tau)$ compared with stochastic sampling, thereby yielding a sample sequence that reflects transition probabilities more faithfully. A proof of boundedness and $O(T^{-1})$ convergence, inherited from the original herding system, is given in Section A.

Additionally, we introduce a delayed-switching mechanism with a margin $\delta > 0$ to mitigate excessive switching of the discrete state variable, by modifying Eq. (9) as follows:

$$\boldsymbol{x}_{t-1} = \underset{\boldsymbol{x} \in \mathcal{V}}{\arg\max}\, (\boldsymbol{w}_t + \boldsymbol{p}_{t-1} + \delta \boldsymbol{x}_t)^\top \boldsymbol{x}. \tag{11}$$

That is, the sample $\boldsymbol{x}_{t-1}$ remains the same as $\boldsymbol{x}_t$ unless another candidate exceeds the objective value by at least $\delta$.

This herding-based denoising algorithm is entirely deterministic; the only source of randomness arises from the initialization of $(\boldsymbol{x}_T, \boldsymbol{w}_T)$, which is chosen randomly from the uniform distribution on $\mathcal{V} \times [0, 1]^K$. By iterating the update rules, the algorithm induces a mapping from a pure noise $(\boldsymbol{x}_T^{(1:L)}, \boldsymbol{w}_T^{(1:L)})$ to a sample $(\boldsymbol{x}_0^{(1:L)}, \boldsymbol{w}_0^{(1:L)})$, which constitutes a piecewise isometry. Unlike the mapping defined by ODE flows in continuous diffusion models, it is not one-to-one and the probability mass is preserved piecewise. In other words, small perturbations in the weight vector persist unless they alter the argmax; however, once the generated sample changes, they cause a drastic shift in the trajectory of the dynamics, reflecting the weakly chaotic behavior with zero Lyapunov exponents almost everywhere.

The time-dependent herding system introduced here can be regarded as a natural extension of the $\Delta\Sigma$-modulator (Inose et al., 1962; Inose & Yasuda, 1963) and the error diffusion algorithm (Adler et al., 2003), which have long been studied in engineering fields such as signal and image processing.

## 4 EXPERIMENTS

We demonstrate the effectiveness of the proposed discrete denoising algorithm through experiments on text and image generation. For clarity of presentation, we reproduce the trained UDLM model as reported in (Schiff et al., 2025) and evaluate the improvements that result solely from introducing the herding algorithm into the reverse denoising process. All experiments are conducted based on the official code provided by Schiff et al. (2025).

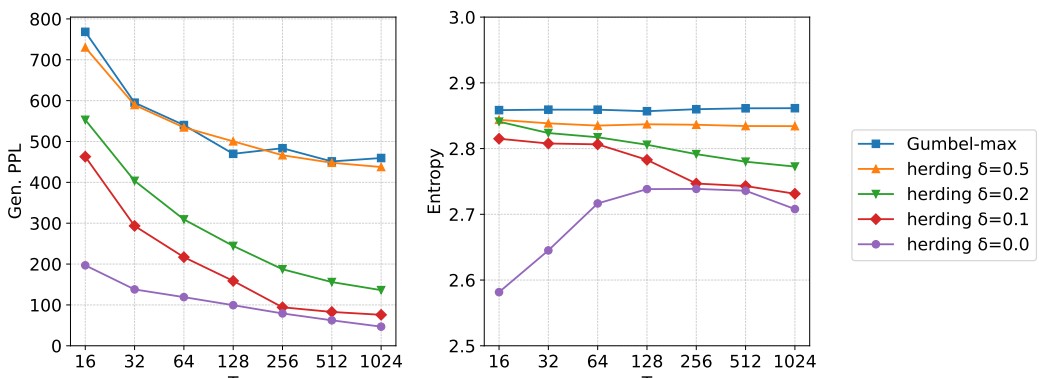

Figure 1: Evaluation of text8 character-level text samples by generative perplexity (Gen. PPL) and entropy. Results for stochastic denoising (Gumbel-max) and herding-based denoising with different $\delta$ values are shown.

### 4.1 CHARACTER-LEVEL TEXT GENERATION

We first evaluate the proposed algorithm in character-level text generation, using the text8 dataset (Mahoney, 2011). The vocabulary size is $35$ and the output length is fixed to $256$ characters. The model architecture follows the UDLM paper (Schiff et al., 2025), which is based on Transformer with 92.4M parameters. For evaluation, we generate 1,024 samples for each condition and compute the generative perplexity based on the GPT-2 Large pretrained model (Radford et al., 2019) and the entropy.

Figure 1 shows the generative perplexity and entropy of the generated samples. The herding-based denoising with $\delta = 0$ consistently outperforms stochastic denoising in terms of perplexity, achieving up to a $10\times$ improvement for $T = 1024$ inference steps, albeit at the expense of entropy. Increasing the margin $\delta$ interpolates between these two extremes, allowing us to improve perplexity with a moderate trade-off in entropy.

### 4.2 WORD-LEVEL TEXT GENERATION

We also evaluate the proposed algorithm in word-level text generation, using the one billion words dataset (LM1B) (Chelba et al., 2013). The vocabulary size is $30,522$ and the output length is $128$ tokens. The model architecture follows the UDLM paper (Schiff et al., 2025), which is based on Transformer with 139M parameters. For evaluation, we generate 1,024 samples for each condition and compute the generative perplexity and MAUVE score (Pillutla et al., 2021) based on the GPT-2 Large pretrained model (Radford et al., 2019) as well as the entropy.

Table 1 shows the evaluation results. In our reproduction experiment, the UDLM model trained using wrapped LM1B sequences performs worse than the published results (Schiff et al., 2025). Nevertheless, the herding-based denoising substantially reduces the perplexity compared to stochastic denoising for both $T = 1024$ and $T = 128$ steps, outperforming the published UDLM (uniform) and MDLM (masked) results and even the autoregressive (AR) model. We also observe a slight loss in entropy and a small improvement in the MAUVE score. Overall, the herding-based denoising significantly improves the coherence and linguistic quality of the generated text while affecting the distributional metrics only minimally. It should also be noted that the generative perplexity improves as the number of inference steps $T$ increases, which suggests that iterative refinement is enhanced by deterministic denoising.

### 4.3 CATEGORICAL-VALUED IMAGE GENERATION

We apply our herding-based algorithm to image generation, following the experiments of D3PM (Austin et al., 2021) and UDLM (Schiff et al., 2025). While pixel values are ordinal by nature, they are treated as discrete categorical variables for the purpose of evaluating generative performance.

Table 1: Evaluation of LM1B text samples by generative perplexity (Gen. PPL), entropy, and MAUVE score. The bottom two rows show our results using the UDLM model trained with the code from Schiff et al. (2025). The herding margin is set to $\delta = 0.0005$ for $T = 1024$ and $\delta = 0.005$ for $T = 128$. Best Gen. PPL values are indicated in bold. [†]Published values from Schiff et al. (2025).

| Method | $T = 1024$ | | | $T = 128$ | | |
| | Gen. PPL ($\downarrow$) | Entropy ($\uparrow$) | MAUVE ($\uparrow$) | Gen. PPL ($\downarrow$) | Entropy ($\uparrow$) | MAUVE ($\uparrow$) |
|---|---|---|---|---|---|---|
| AR[†] | – | – | – | 67.46 | – | – |
| MDLM[†] | 116.80 | – | – | 120.93 | – | – |
| UDLM[†] | 78.22 | – | – | 79.87 | – | – |
| UDLM (reproduced) | 98.75 | 6.83 | 0.0116 | 98.33 | 6.81 | 0.0119 |
| UDLM herding (ours) | **46.71** | 6.30 | 0.0136 | **60.39** | 6.34 | 0.0135 |

Table 2: Evaluation of CIFAR-10 image samples by FID and IS. The bottom two rows show our results using the UDLM models trained with the code from Schiff et al. (2025). D-CFG denotes conditional models with $\gamma = 1$. The herding margin is set to $\delta = 0.07$ for $T = 128$ and $\delta = 0.15$ for the other cases. Best values are indicated in bold. [†]Published values from Austin et al. (2021) and Schiff et al. (2025).

| Method | $T = 1000$ | | $T = 1024$ (D-CFG) | | $T = 128$ (D-CFG) | |
| | FID ($\downarrow$) | IS ($\uparrow$) | FID ($\downarrow$) | IS ($\uparrow$) | FID ($\downarrow$) | IS ($\uparrow$) |
|---|---|---|---|---|---|---|
| D3PM Uniform[†] | 51.27 | 5.99 | – | – | – | – |
| MDLM[†] | 33.75 | 6.74 | 27.94 | 7.14 | 64.09 | 5.81 |
| UDLM[†] | 33.65 | 6.86 | 26.70 | 7.43 | 30.48 | 7.30 |
| UDLM (reproduced) | 32.64 | 6.97 | 24.94 | 7.41 | 29.19 | 7.18 |
| UDLM herding (ours) | **26.39** | **7.40** | **19.20** | **7.79** | **23.95** | **7.51** |

Namely, the forward uniform noising process assigns equal probability to transitions to neighboring and distant pixel values alike.

The dataset is discretized CIFAR-10 (Krizhevsky & Hinton, 2009), where the pixel intensities of the $32 \times 32$ RGB images are discretized to 8-bit integer values. The model is based on a U-Net architecture (Ronneberger et al., 2015), which is identical to that used in Schiff et al. (2025). For evaluation, we computed Fréchet inception distance (FID) (Heusel et al., 2017) and inception score (IS) (Salimans et al., 2016) on 50,000 generated samples, using the PyTorch evaluation code [1] with a pretrained Inception v3 model (Szegedy et al., 2016).

From the evaluation results shown in Table 2, we confirm that the trained unconditional UDLM model and conditional UDLM model (D-CFG with $\gamma = 1$) reproduce the published values nearly identically. The proposed herding-based denoising consistently outperforms stochastic denoising in both FID and IS across all cases. Remarkably, the herding-based denoising with $T = 128$ steps achieves performance comparable to that of stochastic denoising with $T = 1024$ steps.

## 5 DISCUSSION

The deterministic denoising algorithm defines a mapping from the initial states $(\boldsymbol{x}_T^{(1:L)}, \boldsymbol{w}_T^{(1:L)})$ to their final states $(\boldsymbol{x}_0^{(1:L)}, \boldsymbol{w}_0^{(1:L)})$, which induces a flow of probability mass within the discrete state space $\mathcal{V}^L$. Each initial state encodes a generated sample $\boldsymbol{x}_0^{(1:L)}$, whose required precision is determined by the granularity of the piecewise isometric mapping. As our approach derandomizes Markov chains, the probability flow can be guided by the transition probabilities, as demonstrated in the numerical experiments with the conditional UDLM model (D-CFG) on the CIFAR-10 dataset. Our algorithm could also be combined with existing inference-time scaling methods (Kit et al., 2025; Singhal et al., 2025; Pani et al., 2025) by updating the particles deterministically.

---

[1]Available at `https://github.com/w86763777/pytorch-image-generation-metrics`

The initial weight vectors $\boldsymbol{w}_T$ are currently chosen randomly from the uniform distribution on $[0,1]^K$, which is a natural starting point as explained in Section A. Scaling the initial weights can influence the balance between the determinism and randomness of the algorithm. Moreover, there is no reason that all time steps should be treated equally in the accumulation of the discrepancy between the empirical and the target distributions. It is natural to consider weighted accumulation, where the weight scheduling can be an important factor in denoising performance.

We found that introducing delayed switching in the herding algorithm is effective in improving the quality of generated samples. However, it is not clear how to set the appropriate switching margin $\delta$, which can depend on the model, the dataset, the vocabulary size $K$, and the number of inference steps $T$. Alternative mechanisms for delayed switching are also possible. Further theoretical and numerical studies will be necessary to clarify the role of delayed switching in deterministic denoising.

These observations on weighted accumulation and delayed switching naturally lead to the idea of continuous-time formulation. Discrete diffusion models formulated as continuous-time Markov chains (CTMCs) have been studied extensively (Campbell et al., 2022; Lou et al., 2024), and can also be naturally derandomized using the continuous-time variant of the herding system, which exhibits chaotic billiard dynamics (Suzuki, 2015). As an alternative, chaotic billiard sampling (Suzuki et al., 2013; Lee et al., 2025), which can be extended to multi-state spin models such as the Potts model (Suzuki, 2013), can also be applied. Compared with the CTMC formulation, these derandomized billiard systems require computing the exact state transition times induced by collisions with the billiard boundary, which is more time-consuming. Since the exact computation may not contribute to the performance, time-discretized computation is considered more practical. Nonetheless, continuous-time formulations may offer advantages in dedicated hardware implementations, as demonstrated in the integrated-circuit implementation of chaotic Boltzmann machines (Yamaguchi et al., 2019). This research direction is related to acceleration methods for the reverse processes of CTMCs (Park et al., 2025; Ren et al., 2025b).

As mentioned in the introduction, we focused on the uniform diffusion process in this paper. However, the proposed herding-based denoising algorithm can also be applied to other Markov-chain formulations of discrete diffusion models. While in Chen et al. (2024) the state transition time from the absorbing state is externally specified, the proposed deterministic algorithm may allow the transition time to be determined by the system itself according to the transition probability, which is conceptually related to an adaptive sampler for masked diffusion (Ben-Hamu et al., 2025). As the remasking mechanism, as employed in LLaDA (Nie et al., 2025; Zhu et al., 2025), makes the reverse process of masked diffusion more dynamic and enables iterative refinement, our approach could be applied to derandomize the remasking process to further enhance the refinement.

The capability of the herding algorithm is not limited to sampling from categorical distributions. By appropriately designing feature functions, additional constraints, such as symmetry, can be incorporated into the generated samples, which may further enhance the applicability of discrete diffusion models. Although we presented a fully deterministic algorithm, it is also possible to combine the method with probabilistic sampling. Since the boundedness of the weight vector is essential, monitoring its dynamics may help improve the performance of stochastic denoising, as in bounded-error sampling (Yamashita & Suzuki, 2019).

In summary, we presented a novel strategy for derandomizing the reverse denoising process of discrete diffusion models, based on the herding algorithm with weakly chaotic dynamics. Our results provide a positive answer to the key question of whether deterministic denoising can also be effective in discrete diffusion models. As our proposed algorithm is minimal, there are many directions for further improvement. We expect that our approach will contribute to enhancing the significance of discrete diffusion in generative modeling.

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

## A    BOUNDED DYNAMICS OF HERDING-BASED DENOISING ALGORITHM

As a theoretical justification of our proposed algorithm in Section 3, we provide a proof of the boundedness of weight vector $\boldsymbol{w}_t$ and the $O(T^{-1})$ convergence of empirical sample distribution to the averaged probability vector.

Let $w_{t,i}$, $p_{t,i}$, $x_{t,i}$ denote the $i$th component of $\boldsymbol{w}_t$, $\boldsymbol{p}_t$, $\boldsymbol{x}_t$, respectively. We first observe that the total weight $\sum_i w_{t,i}$ is conserved, as

$$\sum_i w_{t-1,i} = \sum_i (w_{t,i} + p_{t-1,i} - x_{t-1,i}) = \sum_i w_{t,i} + 1 - 1 = \sum_i w_{t,i}.$$

Let $\mu = \sum_i w_{t,i}/K$ be the average weight. The initial weight satisfies $w_{T,i} \geq \mu - 1$ because $w_{T,i}$ is chosen from $[0, 1]$. Now assume that $w_{t,i} \geq \mu - 1$ holds for some $t$. If $x_{t-1,i} = 0$, then $w_{t-1,i} = w_{t,i} + p_{t-1,i} - x_{t-1,i} \geq w_{t,i} \geq \mu - 1$. If $x_{t-1,i} = 1$, then $w_{t,i} + p_{t-1,i}$ attains the maximum value, which is at least the average $\mu$. Hence, $w_{t-1,i} = w_{t,i} + p_{t-1,i} - x_{t-1,i} \geq \mu - 1$. Thus, in both cases, we obtain $w_{t-1,i} \geq \mu - 1$, and by induction the inequality holds for all $t$. This establishes a uniform lower bound for each component of $w_t$. Since a lower bound exists and the total weight $\sum_i w_{t,i}$ is conserved, every component of $\boldsymbol{w}_t$ must remain bounded from above as well, and therefore $\boldsymbol{w}_t$ is bounded. The same argument applies to the delayed-switching version, in which case the lower bound becomes $\mu - 1 - \delta$.

The $O(T^{-1})$ convergence follows from the boundedness of $\boldsymbol{w}_t$, as

$$\frac{1}{T} \sum_{\tau=0}^{T-1} (\boldsymbol{p}_\tau - \boldsymbol{x}_\tau) = \frac{1}{T} \sum_{\tau=0}^{T-1} (\boldsymbol{w}_\tau - \boldsymbol{w}_{\tau+1}) = \frac{1}{T} (\boldsymbol{w}_0 - \boldsymbol{w}_T) \to 0 \qquad (T \to \infty).$$

For initialization, it is natural to choose $\mathbf{w}_T$ randomly from the uniform distribution on $[0, 1]^K$ for the following reason. The stationary distribution of the uniform-diffusion Markov chain is the uniform distribution over the sample space $\mathcal{V}$. Therefore, we regard $\boldsymbol{w}_T$ as a weight vector obtained by iterating the herding updates for the uniform probability vector $\boldsymbol{p}_t = \mathbf{1}/K$ for all $t \geq T$, as if the updates had continued from $t = \infty$. Then, after a transient period, each weight $w_{t,i}$ increases by $1/K$ at each iteration but decreases by 1 once every $K$ iterations. As a result, $w_{T,i}$ stays within an interval of length 1. Hence, we choose the uniform distribution on $[0, 1]^K$ as a natural starting point.

## B    RANDOM TEXT SAMPLES

Below we present the first eight outputs for each of stochastic and deterministic denoising, generated by the same UDLM model trained on the LM1B dataset with $T = 1024$ steps (Table 1).

### B.1    STOCHASTIC DENOISING

*[CLS] lukewarm to crash the world's \$ 3 billion ( £1. 4 billion ) party. [CLS] it has helped to quench anger of the regular customers who long ago bought exclusive movies on the web or through itunes. [CLS] nokia has several partnerships with manufacturers to prevent the mobile phone markets from dying and plans to market several models to only the end sellers by 2010. [CLS] " i am addicted to pain relief. [CLS] do - - vote for app print / audio. [CLS] john s. senior of the university of washington center, seattle, predicted that the previous projection of 2. 01 million jobs this year were created in 2006. [CLS]*

*[CLS] suffering or the public interest, " the minister told reporters. [CLS] in a rare move in mid - eastern virginia, stevens edged him out of warm washington politics. [CLS] between celebrating with a journey to boot the pumas have become victims of a new year of joy. [CLS] san diego fell to 6 - 0 in 25 of the 22, 000 seats at azteca stadium. [CLS] by the time they had arrived at 3. 9am a riot ensued to which no one would respond. [CLS] scandinavian investors cannot replace workers after their boards lose more than 6, 500 heads every week to other large single ventures, light of day, and today [CLS]*

*[CLS]iawski, i koch ( a barquierja ), steven, ( espanola aurth ), gilbert, juan jean - baptiste ( fyzi kosian ), dujellschhofer, salim he - nazarzi, korbowu, gousset i yves heymans, pierre glover, isaiah, givat, beauxmele, plygielewski, racec, gedizzola, camoranesi, m bijzyrafors, b czemolo. [CLS] even so, rain began in the morning, left the field and put a [CLS]*

*[CLS] " it was inevitably here. [CLS] dr. h. richard hurley of the university of texas injected alexander with a tiny drug histoxipturably k to repair stuttering muscles ahead of his procedure. [CLS] face the press, so why's more? [CLS] as an outsider, oussi is still led by his armed yet steady partner. [CLS] cano drilled a solo homer in the 11th, striking the wrist of rodriguez to juice abdullah's duel with adam jones, the other in season 3 debut. [CLS] meteorologist rossio said as they tested the lake floor, temperatures found themselves hot, reaching sixc. [CLS] [CLS]*

*[CLS]. [CLS] geithner also exaggerates her comments on leading the united states out of a financial crisis that has had a huge negative impact on the nation's economy, and is on the currency's creation as well as on other nations that have given power to their governments. [CLS] but their response demonstrated weakly that others might preach about direness, and some looked appropriate. [CLS] boiona de gomez of the left ( left ) did the work until her son lourdes, and son hilda of salvador, both grew up. [CLS] avery has had his tools running through 16 - inning bats and will have 385 at - bats [CLS]*

*[CLS] the department of justice ; any of the actions needed may be called for the suspension of operations of any portion of the above guarana series due ( notes terminate or revoke the note ) or be required to transport the vehicle in need ; public web site resources available at website. usw. gov / aftol / increasest and ldcpp for specific. [CLS] how will his dog mane affect people? [CLS] nasa engineers said the name has been up since the 2004, considering an expected number of launches last year from the 2001. [CLS] it is also the way the latest compilation is a mistake. [CLS] christakis, [CLS]*

*[CLS] brownback partners are the final participants in a half - day meeting. [CLS] unlike avid house - hunters who are more inclined to take videos of quality gadgets like web mani on toys. com, mr o'brien says regular updates on controversial tools or full encompasses of predators and a multitude of natural disasters dog other online markets. [CLS] dawn brancheau had explicitly cycled into the river seine with the remains of her dead husband. [CLS] i wonder why we do not run older companies rootcare problems. [CLS] the report showed americans are still split over u. s. economy's slow pace, deflation levels and unemployment. [CLS] a [CLS]*

*[CLS] the discussion. - consolidatedtable conclusions both as a historical and comprehensive transaction. sales, revenue and revenues for moving forward in one empty but titled year. been in filings on form s - 8b filed by the sec with the securities and exchange. [CLS] they are normally spent during trials at the court itself. [CLS] djokovic quickly began his resurgence, winning an emphatic ninth game as federer hauled down three break points. [CLS] organised crime, the always, streaked this true. [CLS] " all cities will benefit, " said niamh myung norungon of the university of california, san diego and colleagues at the [CLS]*

## B.2   DETERMINISTIC DENOISING

*[CLS] renault - nissan alliance, which has been with the white house. [CLS] so check the scrape to tell you what you can... you can. [CLS] the first one you saw, that was called " the doctrine of the doctrine of the faith, what right? " [CLS] who is in damage control? [CLS] " no one wants to shoot at a building next to the supreme court, " he said. [CLS] this has been the only long - running issue - that we turned up for the t20 world championship instead of ashes series and i'm sure that it's going to be the number two decision. [CLS] as the [CLS]*

*[CLS] get there, i won't pass it up, but i would prefer to be in the country in 2009, " he said. [CLS] so what does this new policy do? [CLS] not even bad, but obama, who had just phoned bush to say it was real, had no need lose himself in a politician's badge of honor. [CLS] the young man's years of abuse started at an age in 1979 when he started praying at the same mosque in birmingham. [CLS] a force spokeswoman said : " while levels of crime at the premises have fallen, it is our responsibility to make sure we can continue these [CLS]*

*[CLS], she said she and her husband will spend time together after the game in cleveland. [CLS] " we are in progress with confidence, with the strength of our 1752 portfolio and the ability to meet our obligations to the market, " he said. [CLS] all five of the banking groups have promised support for a scheme worth more than £50 billion, 69 % of which was borrowed before the tax change in april. [CLS] his father told the priest that the boy would come out of retirement but would only return home at the end of his life. [CLS] he was also involved in the crash. [CLS] if you happen to wait for the day, [CLS]*

702
703
704
705
706
707
708

*[CLS] is any way to stop, speak to me. [CLS] it said that it was channeling shareholders through a " period of difficult trading " that had been far short of expectations. [CLS] " access becomes an issue for the children every time we get out of the school, " the spokesman said. [CLS] " he really is never going to settle down. [CLS] she was the proud parent of three of her children who said it was a nightmare. [CLS] davis said bush's words helped ease her train ride. [CLS] samples of the water, thought to have been drained due to sewage, would be available in the area over three weeks. [CLS] [CLS]*

709
710
711
712
713

*[CLS] near - constant tension in darfur, one of the world's poorest, about 60 miles east of khartoum. [CLS] taken from the gses' perspective, " the urban unemployment rate has dropped 30 percent in the past year, not by more than 7 percent, but only by 5 percent, " the group wrote in the original report, which favored health, safety and financial standards. [CLS] " i'd like you to, " he said. [CLS] " we have the power, one agency at a time... respond to a new law, " he assured congress. [CLS] o. k. the axle [CLS]*

714
715
716
717
718
719
720

*[CLS] battle of her own in the us election for the nomination. [CLS] also, officials say 911 callers confirm that the caller is related to a suspect in the vicinity. [CLS] there is no immediate prospect that fiat motors, a large u. s. - owned subsidiary that competes with the italian automaker fiat, will take over operations of its michigan plant, gm said. [CLS] former lover michelle roberts, 31, called the matter a long - distance affair. [CLS] and since foreign troops came to their assistance early in 2007, the afghan authorities have faced a growing flood of taliban military forces. [CLS] it offers a lot of features, " said [CLS]*

721
722
723
724
725
726

*[CLS] [CLS] and she says : i don't think she's very lucky but today she's feeling better. [CLS] prices are still falling. [CLS] this would be microsoft's biggest buy - out exercise ever in yahoo's long history, but executives say microsoft could have time to turn it around in just over a year. [CLS] just when the war was under way, u. s. forces were in iraq for the entire year and spent nearly four times the u. s. budget for operations in their adopted country in 2003, 2004 and october 2007. [CLS] it also includes the vast majority of the big east's major [CLS]*

727
728
729
730
731
732
733

*[CLS] today's economy, new schools and more flexibility, " duncan said. [CLS] and, yes, we were confused. [CLS] the next chapter represents an important move toward the right. [CLS] and as it stands today, i hesitate to say this because the administration still has a specific intent to put the obama administration's unemployment rate at 7 million to 9. 7 million. [CLS] comment : some times he was able to take an on - line gender issue for political gain... maybe he was just asking why. [CLS] you know what, we must change if we all want to understand this. [CLS] " there is no time for [CLS]*
