# OpenReview forum: "Deterministic Discrete Denoising"
_ICLR.cc/2026/Conference — Submitted to ICLR 2026_

### Official Review · Reviewer_B8Lv · 2025-10-17

**Soundness:** 2
**Presentation:** 1
**Contribution:** 2
**Rating:** 2
**Confidence:** 3

**Summary:**

This paper introduces a deterministic algorithm for discrete diffusion models. The proposed method is inspired by the herding algorithm with weakly chaotic dynamics. It is claimed that the proposed method achieves better efficiency and sample quality on both text and image generation tasks.

**Strengths:**

- The idea of deterministic sampling for discrete diffusion models is very interesting and relevant.
- The usage of the herding algorithm seems plausible.
- Experiments showed promising results.

**Weaknesses:**

- The paper is written in an extremely compact and concise manner. This makes many arguments unclear and hard to read.
- The proposed herding algorithms lack explanation and justification. To me, it is rather unclear how the scheme in Eqns. 7 and 8 can be directly modified to Eqns. 9 and 10.

See more in Questions.

**Questions:**

- My foremost question is that while the herding algorithm seems iterative, the proposed denoising alternative (Eqns. 9 and 10) is not iterative. To the best of my understanding, the reason the herding algorithm works is the convergence in Eqn. 6, so that given any target probaility vector $p$, we can construct a herding dynamics that produces samples whose distribution converges to $p$ as $T \to \infty$. However, the dynamics in Eqns. 9 and 10 only produces one set of $x_0$ and $w_0$. I just wonder if this could guarantee that the generation is unbiased.
- Could you please elaborate more on the delayed-switching mechanism? It is only mentioned in a single sentence but seems matter a lot to the final performance in experiments.

---

> ### Author Response · Authors · 2025-11-24
> **Response to Reviewer B8Lv**
>
> Thank you very much for your critical review and for your positive evaluation of our basic idea.
>
> Regarding your important questions on the time-dependent version and delayed switching, we provide a detailed explanation in a separate comment [here](https://openreview.net/forum?id=CHtLFyDbZp&noteId=vszATw0NNM), which we hope addresses your concerns.

---

### Official Review · Reviewer_hA6i · 2025-10-21

**Soundness:** 2
**Presentation:** 3
**Contribution:** 3
**Rating:** 2
**Confidence:** 4

**Summary:**

This paper tackles the important problem of deterministic denoising in discrete diffusion via the classical herding algorithm from MCMC. The paper introduces this algorithm in the context of denoising diffusion, along with some modifications. This algorithm does not requires any training, and can work with any discrete diffusion model with uniform noising out-of-the-box. Experiments on text and image generation are presented.

**Strengths:**

The setting considered in the paper is very important and timely. The algorithmic ideas from classical MCMC literature can be very useful in this context. The paper explains the origin of the algorithm well, with multiple connections being made to this literature. I personally think that this line of reasoning can be very impactful. The experiments show some improvements in image generation experiments and in generative perplexity in text experiments.

**Weaknesses:**

- The authors do not explain many technical concepts well. For instance, the paper talks about weakly chaotic behavior, Delta-Sigma etc without much explanation. What does chattering mean (Line 172)?  A short primer should be provided, at least in the appendix. There is ample space in the main paper itself. I understand that some of these concepts can be common knowledge in a different research community, however care has to be taken to explain these appropriately to the generative modeling community.

- An appendix with detailed explanation is missing no pseudo code and detailed algorithmic description. For instance, delayed switching is explained in words, but a detailed technically sound exposition is not provided. The reason the time dependent version in Section 3 works is not well explained.

- Generated Samples are not provided. This is necessary for both image and text generation experiments as a "vibe check".

- In the text experiments on LM1B, while the generative perplexity decreases with respect to the baseline, entropy also decreasing is generally not a good thing.  Generative perplexity alone cannot be regarded as a good metric. Sequences such as “.......like like like…..” achieve very low generative perplexity. I suggest evaluation using the MAUVE score to obtain a full picture.

As such I do not consider this paper to be ready for publication, despite the very promising direction.

**Questions:**

Please address my points in the weaknesses section.

---

> ### Author Response · Authors · 2025-11-24
> **Response to Reviewer hA6i**
>
> We appreciate your positive evaluation of our research direction and your important suggestions for improving the clarity of our contributions. We will reflect your comments in future versions of our manuscript.
>
> Regarding the time-dependent version and delayed switching, we provide a detailed explanation in a separate comment [here](https://openreview.net/forum?id=CHtLFyDbZp&noteId=vszATw0NNM).
> By “chattering,” we intended to refer to excessive switching between selected samples. Such "chattering" is common in mechanical switches and is typically mitigated by introducing a hysteresis margin, which motivated us to introduce delayed switching. We hoped this analogy would help convey the intuitive motivation, but we now realize that it was an inappropriate choice of terminology.
>
> As we updated the text experiments on LM1B, we provide the random output samples for stochastic and deterministic denoising in separate comments [here](https://openreview.net/forum?id=CHtLFyDbZp&noteId=WAgptH40QQ).

---

### Official Review · Reviewer_uEJ8 · 2025-10-25

**Soundness:** 2
**Presentation:** 3
**Contribution:** 2
**Rating:** 2
**Confidence:** 4

**Summary:**

This paper proposed a deterministic algorithm for sampling from a uniform discrete diffusion model, in the sense that the only source of randomness is the initial random noise. The algorithm is based on the variant of herding algorithm. The authors demonstrated the performance of the proposed method on text and image generation tasks.

**Strengths:**

To the best of my knowledge, the herding algorithm proposed in this paper has not been applied to the task of sampling from discrete diffusion models before, which makes this work original. As shown in their experiments, the proposed method achieves competitive performance compared to existing stochastic sampling algorithms. The paper is well-written and clearly structured. The mathematical notations are mostly clean and consistent.

**Weaknesses:**

The largest weakness of the paper lies in its limited scope of applicability. The proposed method is only effective for **uniform** discrete diffusion models, and the experiments are also all limited to this setting. However, the current trend in discrete diffusion models is to use **masked** discrete diffusion models, which achieve better performance in various tasks, especially in text generation, and are scalable to larger models and datasets. Due to the nature of the masked discrete diffusion models, the proposed method cannot be directly applied, so I feel the impact of the proposed method is limited.

Moreover, as the uniform discrete diffusion models do not perform well on many of the modalities than masked discrete diffusion models or other benchmarks, the improvements shown in the experiments may not be very meaningful in practice. The paper only demonstrates quantitative improvements in terms of generative perplexity, entropy, FID and IS. These seemingly significant improvements do not necessarily translate to better quality of generated samples, as they can be as bad as the one produced by stochastic sampling methods on the same models. For instance, 74.95 perplexity on GPT-2 level text may still be of low quality, and 19.20 FID on CIFAR-10 is much worse than most of the state-of-the-art generative models nowadays, which have FID scores below 10. It would be more convincing if the authors could provide qualitative results of the generated samples.

In addition, I understand it is an interesting theoretical problem, but it would be beneficial if the authors could provide more convincing reasons on the motivation of studying deterministic samplers. For continuous diffusion models, deterministic ODE samplers are preferred as their generated trajectories are straighter ($C^1$ continuity of ODE v.s. $C^{\frac12-}$ continuity of SDE), and thus lead to smaller discretization errors when using the same step size. Moreover, deterministic ODE samplers enable distillation (e.g., consistency model). However, I don't see similar motivations for discrete diffusion models.

Overall, I think the paper provides an interesting algorithm, but its limited scope of applicability makes it less impactful in practice. While the work contains some novel ideas, the combination of limited applicability, modest experimental results, and lack of strong theoretical foundations suggests it doesn't meet the standards for top-tier ML/AI conferences.

**Questions:**

Could the authors provide some theoretical guarantees on the convergence of Eqs. (7,8)? Can we prove that $||\mathbf{w} _ t||$ is bounded under some mild assumptions in this simple case? As the authors have mentioned in Sec. 5, the initialization of $\mathbf{w} _ T$ is $\text{Unif}([0,1]^K)$. How does this choice affect the convergence, and is there any reason why this specific initialization was chosen, instead of, say, $\text{Unif}([-1,1]^K)$?

Comparing Eqs. (7,8) and Eqs. (9,10), why there's $(\mathbf{w} _ t+\mathbf{p} _ {t-1})^\top\mathbf{x}$ instead of $\mathbf{w} _ t^\top\mathbf{x}$? Is there any intuition of the convergence and boundedness of the weights in this time-inhomogeneous case? Also, as in practice we consider the sequence of $L$ tokens, i.e., $\mathcal{V}^L$ instead of $\mathcal{V}$, the notations in Eqs. (9,10) need to be modified accordingly. Pseudo-code may also be helpful to present the algorithms in a clearer way.

As pointed out in the weaknesses, I suggest the authors to provide some qualitative results of the experiments. Also, a more detailed discussion on the related works on herding algorithms would be helpful to position the proposed method in the literature. In particular, more explanations on UDLM is needed to for readers unfamiliar with it.

---

> ### Author Response · Authors · 2025-11-24
> **Response to Reviewer uEJ8**
>
> Thank you very much for your thorough review,  your important suggestions, and your positive  evaluation of the novelty and clarity of our work.
>
> We fully agree that masked diffusion models are a major current trend. At the same time, it is important to consider broader directions for future developments. One of the key benefits of discrete diffusion models over autoregressive models is iterative refinement, which can be enhanced by making the reverse process more dynamic. In our manuscript, we demonstrated the effectiveness of our approach on the uniform diffusion model, which serves as the simplest playground with a dynamic reverse process. However, we do not intend to restrict our future applications to uniform diffusion. Indeed, the remasking mechanism in LLaDA highlights the importance of iterative refinement in masked diffusion, and our proposed deterministic denoising may also have the potential to contribute to this trend toward more dynamic reverse processes, as mentioned in our reply to Reviewer ezsi. Although our aim is to demonstrate the effectiveness of derandomizing the reverse process of existing models, the experimental results (updated Table 1 and original Table 2) show that stochastic and deterministic denoising of uniform diffusion (UDLM) consistently outperform masked diffusion (MDLM). While we recognize that masked diffusion models are rapidly advancing, we consider it equally important to continue exploring broad directions including uniform diffusion.
>
> We also thank you very much for your comments on the motivation behind our approach. The primary motivation for considering a deterministic herding-based sampler is its $O(T^{-1})$ convergence rate, which is clearly better than the $O(T^{-1/2})$ rate of stochastic samplers. A proof of this convergence rate is provided in a separate comment. The herding updates can be viewed as representing a sequence of continuous probability vectors using discrete samples, where the approximation error is reduced by derandomization. We see this as analogous to the continuity of deterministic trajectories and the resulting reduction of discretization error in continuous diffusion models. Moreover, several advantages of deterministic samplers in continuous models, such as the ability to encode samples through the choice of initial values, are also retained in the discrete setting. In a broader sense, we raise a fundamental question of whether stochastic algorithms are necessary at all in the discrete-state reverse process.
>
> > Could the authors provide some theoretical guarantees on the convergence of Eqs. (7,8)? Can we prove that $||\mathbf{w} _ t||$ is bounded under some mild assumptions in this simple case?
>
> Yes, we provide a proof for a more general case in a separate comment as described below.
>
> > As the authors have mentioned in Sec. 5, the initialization of $\mathbf{w} _ T$ is $\text{Unif}([0,1]^K)$. How does this choice affect the convergence, and is there any reason why this specific initialization was chosen, instead of, say, $\text{Unif}([-1,1]^K)$?
>
> The stationary distribution of uniform diffusion is the uniform distribution over the sample space. Therefore, we consider iterating the herding updates at the fixed time $T$ for the uniform distribution $p_{T,i}=1/K$ for all $i$. After the transient period, $w_{T,i}$ increases by $1/K$ at each iteration but decreases by $1$ once every $K$ iterations. As a result, $w_{T,i}$ remains within an interval of length $1$. This is why we choose the uniform distribution on $[0,1]$ as a natural starting point. Although our convergence proof assumes an interval of length $1$, this assumption can be relaxed by taking the transient dynamics into account.
>
> > Comparing Eqs. (7,8) and Eqs. (9,10), why there's $(\mathbf{w} _ t+\mathbf{p} _ {t-1})^\top\mathbf{x}$ instead of $\mathbf{w} _ t^\top\mathbf{x}$? Is there any intuition of the convergence and boundedness of the weights in this time-inhomogeneous case?
>
> We are sorry that we did not describe these points clearly. In this [comment](https://openreview.net/forum?id=CHtLFyDbZp&noteId=vszATw0NNM), we provide the motivation as well as concrete proof of convergence and boundedness in time-inhomogeneous case.
>
> > Also, as in practice we consider the sequence of $L$ tokens, i.e., $\mathcal{V}^L$ instead of $\mathcal{V}$, the notations in Eqs. (9,10) need to be modified accordingly. Pseudo-code may also be helpful to present the algorithms in a clearer way.
> > As pointed out in the weaknesses, I suggest the authors to provide some qualitative results of the experiments. Also, a more detailed discussion on the related works on herding algorithms would be helpful to position the proposed method in the literature. In particular, more explanations on UDLM is needed to for readers unfamiliar with it.
>
> Thank you very much for your valuable suggestions to help make our manuscript clearer. We will reflect your comments in future versions of our manuscript.

---

### Official Review · Reviewer_ezsi · 2025-11-02

**Soundness:** 3
**Presentation:** 3
**Contribution:** 2
**Rating:** 4
**Confidence:** 4

**Summary:**

This paper proposed a deterministic algorithm for the denoising procedure in discrete diffusion models, which can be thought of as an analog of Denoising Diffusion Implicit Models (DDIM) for the setting of discrete diffusion models. Specifically, instead of using the categorical sampler or the Gumbel-max trick for each time step in the denoising procedure, the authors borrowed ideas from the herding algorithm and chaotic dynamical systems to introduce a “DDIM-like” deterministic reverse process for discrete state spaces. Numerical experiments on tasks like text and image generation are provided to justify the effectiveness of proposed methodology.

**Strengths:**

(1) This paper studies an important topic with clear motivation, positioning and presentation. To the best of the reviewer's knowledge, there has been no prior work that investigates how deterministic samplers like DDIM can be generalized to the setting of discrete diffusion models.

(2) The reviewer finds the perspective of chaotic dynamical systems/herding algorithm introduced in this paper to be particularly interesting, which has the potential to motivate further studies.

**Weaknesses:**

(1) One possible drawback of the paper is that a theory part seems to be missing here, especially given that the authors only presented two sets of experiments (text and image generation) for the empirical part. Though the reviewer understands that the focus of the manuscript is on the algorithm and methodology, the authors are encouraged to include at least a literature review of existing work on the theoretical properties of discrete diffusion models (A incomplete list of related literature: [1,2,3,4,5,6,7]) and briefly comment on how the proposed algorithm can possibly be analyzed theoretically by combining [1-7] with results derived from previous work on the theoretical properties of the herding algorithm (An incomplete list of related references: [8,9,27]).

(2) Though the experiments conducted here cover both text and image generation, it seems to the reviewer that the quality of the manuscript can be further improved by adding extra baselines. Most importantly, it seems that the authors didn't include [9] as one baseline for the MDLM case, which definitely needs to be added. Moreover, the authors should probably also consider adding a review of related work on existing methods for accelerating the inference speed/improving the generation quality of discrete diffusion models. An incomplete list of related work include but not limit to distillation-based methods [10,11,12,13], design of sampling schedule [14,15], high-order numerical solvers [16] and other techniques [17], etc. Especially for training-free methods like [14,15,16,17], the authors are encouraged to include (a subset of) them as extra baselines for the experiments here.

**Questions:**

(1) Given that large language diffusion models has emerged as a hot topic recently, would it be possible for the authors to comment on whether it would be possible to generalize the methodology proposed here to LLaDA [18,19]? If so, how do the algorithm relate to standard algorithms like parallel/speculative decoding in LLMs?

(2) Since the time-dependent scheme introduced in this paper augments every discrete token with a continuous weight and then performs an argmax-style update at each timestep, would it be possible for the reviewer to ask how does the dynamics in this paper relate to the inference-time scaling framework for diffusion models (an incomplete list of related references: [20,21,22,23,24,25,26]), which all essentially use the Sequential Monte Carlo algorithm to simulate a gradient flow of Fisher–Rao type? Does actually lead to the inherent connection between Fisher-Rao dynamics and the dynamical system associated with the herding algorithm?

Overall, I think the paper might be considered for top ML venues like ICLR, but the authors should probably address all questions above, add papers listed below as extra references and discuss them appropriately.

References:

[1] Srikanth, Aadithya, Mudit Gaur, and Vaneet Aggarwal. "Discrete State Diffusion Models: A Sample Complexity Perspective." arXiv preprint arXiv:2510.10854 (2025).

[2] Huang, Xunpeng, Yingyu Lin, Nishant Jain, Kaibo Wang, Difan Zou, Yian Ma, and Tong Zhang. "On the Complexity Theory of Masked Discrete Diffusion: From $\mathrm {poly}(1/\epsilon) $ to Nearly $\epsilon $-Free." arXiv preprint arXiv:2509.21835 (2025).

[3] Liang, Yuchen, Yingbin Liang, Lifeng Lai, and Ness Shroff. "Discrete Diffusion Models: Novel Analysis and New Sampler Guarantees." arXiv preprint arXiv:2509.16756 (2025).

[4] Liang, Yuchen, Renxiang Huang, Lifeng Lai, Ness Shroff, and Yingbin Liang. "Absorb and Converge: Provable Convergence Guarantee for Absorbing Discrete Diffusion Models." arXiv preprint arXiv:2506.02318 (2025).

[5] Ren, Yinuo, Haoxuan Chen, Grant M. Rotskoff, and Lexing Ying. "How discrete and continuous diffusion meet: Comprehensive analysis of discrete diffusion models via a stochastic integral framework." arXiv preprint arXiv:2410.03601 (2024).

[6] Zhang, Zikun, Zixiang Chen, and Quanquan Gu. "Convergence of score-based discrete diffusion models: A discrete-time analysis." arXiv preprint arXiv:2410.02321 (2024).

[7] Chen, Hongrui, and Lexing Ying. "Convergence analysis of discrete diffusion model: Exact implementation through uniformization." arXiv preprint arXiv:2402.08095 (2024).

[8] Harvey, Nick, and Samira Samadi. "Near-optimal herding." In Conference on Learning Theory, pp. 1165-1182. PMLR, 2014.

[9] Bach, Francis, Simon Lacoste-Julien, and Guillaume Obozinski. "On the equivalence between herding and conditional gradient algorithms." arXiv preprint arXiv:1203.4523 (2012).

[9] Chen, Zixiang, Huizhuo Yuan, Yongqian Li, Yiwen Kou, Junkai Zhang, and Quanquan Gu. "Fast sampling via discrete non-markov diffusion models with predetermined transition time." Advances in Neural Information Processing Systems 37 (2024): 106870-106905.

[10] Hayakawa, Satoshi, Yuhta Takida, Masaaki Imaizumi, Hiromi Wakaki, and Yuki Mitsufuji. "Distillation of discrete diffusion through dimensional correlations." arXiv preprint arXiv:2410.08709 (2024).

[11] Fu, Feiyang, Tongxian Guo, and Zhaoqiang Liu. "Learnable Sampler Distillation for Discrete Diffusion Models." arXiv preprint arXiv:2509.19962 (2025).

[12] Zhu, Yuanzhi, Xi Wang, Stéphane Lathuilière, and Vicky Kalogeiton. "Di [M] o: Distilling masked diffusion models into one-step generator." In Proceedings of the IEEE/CVF International Conference on Computer Vision, pp. 18606-18618. 2025.

[13] Zhu, Yuanzhi, Xi Wang, Stéphane Lathuilière, and Vicky Kalogeiton. "Soft-di [m] o: Improving one-step discrete image generation with soft embeddings." arXiv preprint arXiv:2509.22925 (2025).

[14] Amin, Alan N., Nate Gruver, and Andrew Gordon Wilson. "Why Masking Diffusion Works: Condition on the Jump Schedule for Improved Discrete Diffusion." arXiv preprint arXiv:2506.08316 (2025).

[15] Park, Yong-Hyun, Chieh-Hsin Lai, Satoshi Hayakawa, Yuhta Takida, and Yuki Mitsufuji. "Jump your steps: Optimizing sampling schedule of discrete diffusion models." In The Thirteenth International Conference on Learning Representations. 2024.

[16] Ren, Yinuo, Haoxuan Chen, Yuchen Zhu, Wei Guo, Yongxin Chen, Grant M. Rotskoff, Molei Tao, and Lexing Ying. "Fast solvers for discrete diffusion models: Theory and applications of high-order algorithms." arXiv preprint arXiv:2502.00234 (2025).

[17] Ben-Hamu, Heli, Itai Gat, Daniel Severo, Niklas Nolte, and Brian Karrer. "Accelerated Sampling from Masked Diffusion Models via Entropy Bounded Unmasking." arXiv preprint arXiv:2505.24857 (2025).

[18] Nie, Shen, Fengqi Zhu, Zebin You, Xiaolu Zhang, Jingyang Ou, Jun Hu, Jun Zhou, Yankai Lin, Ji-Rong Wen, and Chongxuan Li. "Large language diffusion models." arXiv preprint arXiv:2502.09992 (2025).

[19] Zhu, Fengqi, Rongzhen Wang, Shen Nie, Xiaolu Zhang, Chunwei Wu, Jun Hu, Jun Zhou et al. "LLaDA 1.5: Variance-Reduced Preference Optimization for Large Language Diffusion Models." arXiv preprint arXiv:2505.19223 (2025).

[20] Singhal, Raghav, Zachary Horvitz, Ryan Teehan, Mengye Ren, Zhou Yu, Kathleen McKeown, and Rajesh Ranganath. "A general framework for inference-time scaling and steering of diffusion models." arXiv preprint arXiv:2501.06848 (2025).

[21] Chen, Haoxuan, Yinuo Ren, Martin Renqiang Min, Lexing Ying, and Zachary Izzo. "Solving inverse problems via diffusion-based priors: An approximation-free ensemble sampling approach." arXiv preprint arXiv:2506.03979 (2025).

[22] Skreta, Marta, Tara Akhound-Sadegh, Viktor Ohanesian, Roberto Bondesan, Alán Aspuru-Guzik, Arnaud Doucet, Rob Brekelmans, Alexander Tong, and Kirill Neklyudov. "Feynman-kac correctors in diffusion: Annealing, guidance, and product of experts." arXiv preprint arXiv:2503.02819 (2025).

[23] Pani, Chinmay, Zijing Ou, and Yingzhen Li. "Test-Time Alignment of Discrete Diffusion Models with Sequential Monte Carlo." arXiv preprint arXiv:2505.22524 (2025).

[24] Ma, Nanye, Shangyuan Tong, Haolin Jia, Hexiang Hu, Yu-Chuan Su, Mingda Zhang, Xuan Yang et al. "Inference-time scaling for diffusion models beyond scaling denoising steps." arXiv preprint arXiv:2501.09732 (2025).

[25] Ren, Yinuo, Wenhao Gao, Lexing Ying, Grant M. Rotskoff, and Jiequn Han. "Driftlite: Lightweight drift control for inference-time scaling of diffusion models." arXiv preprint arXiv:2509.21655 (2025).

[26] Lee, Cheuk Kit, Paul Jeha, Jes Frellsen, Pietro Lio, Michael Samuel Albergo, and Francisco Vargas. "Debiasing guidance for discrete diffusion with sequential monte carlo." arXiv preprint arXiv:2502.06079 (2025).

[27] Lacoste-Julien, Simon, Fredrik Lindsten, and Francis Bach. "Sequential kernel herding: Frank-Wolfe optimization for particle filtering." In Artificial Intelligence and Statistics, pp. 544-552. PMLR, 2015.

---

> ### Author Response · Authors · 2025-11-24
> **Response to Reviewer ezsi**
>
> We appreciate your detailed evaluations and questions with many references, from which we have gained valuable insights.
>
> Regarding Weakness (2), although the experimental settings are not exactly matched, the results reported for DNDM in [9] on the text8 dataset provide a reasonable baseline. In [9], the generative PPL decreases from 1465.75 with a 1000-step masked diffusion model to 600.02 with the 256-step DNDM. In contrast, our method reduces the generative PPL of UDLM from 459.53 with 1024 steps to 79.16 using 256-step herding. While this comparison is limited to the text8 dataset, these numbers indicate that our approach outperforms the results reported for DNDM in both the magnitude of improvement and the achieved generative PPL.
>
> > (1) Given that large language diffusion models has emerged as a hot topic recently, would it be possible for the authors to comment on whether it would be possible to generalize the methodology proposed here to LLaDA [18,19]? If so, how do the algorithm relate to standard algorithms like parallel/speculative decoding in LLMs?
>
> Although LLaDA is formulated as a masked diffusion model, one of its key features is remasking, which enables iterative refinement during generation. This makes the reverse process more dynamic and can be viewed as moving one step closer to uniform diffusion.
> Since our approach has only been demonstrated for uniform diffusion, the direct answer to your question is that a generalization to masked diffusion would be necessary. However, your question suggests a simpler and more interesting research direction: derandomizing the remasking process in LLaDA using our approach, which we believe is worth exploring.
>
> > (2) Since the time-dependent scheme introduced in this paper augments every discrete token with a continuous weight and then performs an argmax-style update at each timestep, would it be possible for the reviewer to ask how does the dynamics in this paper relate to the inference-time scaling framework for diffusion models (an incomplete list of related references: [20,21,22,23,24,25,26]), which all essentially use the Sequential Monte Carlo algorithm to simulate a gradient flow of Fisher–Rao type? Does actually lead to the inherent connection between Fisher-Rao dynamics and the dynamical system associated with the herding algorithm?
>
> Thank you very much for your insightful question. Indeed, each step of our time-dependent scheme induces a flow of probability mass over the sample space, and could possibly be related to or regulated by some gradient flow. However, it is not straightforward to theoretically capture how much mass from the initial uniform distribution is actually transported through the iterations.
> A straightforward answer to your question is that our approach may allow each particle of SMC to be driven deterministically, while requiring a small trick to avoid copies of the same particle resampled multiple times from following exactly the same deterministic trajectory. Your question also led us to recognize another interesting research direction: applying the method in your reference [27] to inference-time scaling, which is aligned with the direction of our work.

---

> > ### Comment · Reviewer_ezsi · 2025-11-25
> > **Response to the authors**
> >
> > The reviewer would like to thank the authors for the detailed rebuttal. Just a gentle reminder that the authors are allowed to revise the manuscript and upload an updated copy during the rebuttal, so it's better to update the paper now (add extra discussions and all references listed above to address all concerns raised in the review). Currently the reviewer is checking concerns raised by other reviewers and will finalize the score soon.

---

> > > ### Author Response · Authors · 2025-11-26
> > >
> > > Thank you very much for your kind advice. We will prepare and upload a revised manuscript soon.

---

### Author Response · Authors · 2025-11-24
**General response**

We sincerely appreciate the highly insightful reviews. Overall, our work aims to introduce the herding algorithm as a bridge to bring ideas from nonlinear dynamics and signal processing into discrete diffusion models. While we have tried to present our ideas using the language of the ML/AI community, we learned a lot from your comments about which aspects should be elaborated and which should be emphasized. Unfortunately, due to computational resource limitations, we found that it was not feasible to conduct new experiments
and address all comments within the discussion period. Nonetheless, with sincere respect for the excellent comments and questions, we would like to respond to each reviewer's questions with what we can currently answer.

In addition, we attach a detailed explanation of the time-dependent herding algorithm, as well as the progress we have made in reproducing the prior work after the submission.

---

> ### Author Response · Authors · 2025-11-24
> **Detailed explanation of time-dependent herding algorithm**
>
> # Time-dependent version and delayed switching
> While the herding algorithm was derived as the zero-temperature limit of energy-based models (Welling, ICML 2009), it can also be interpreted as minimizing the discrepancy between the generated samples and their target, represented by $w_t$.
>
> Accordingly, in the time-dependent version, we choose the maximizer of $(w_t+p_{t-1})^\top x$ in Eq. (9) in order to minimize the maximal component of $w_{t-1}$ in Eq. (10). As Reviewer uEJ8 pointed out, we maximize $(w_t+p_{t-1})^\top x$ instead of $w_t^\top x$. This makes the algorithm greedier with respect to the current probability vector $p_{t-1}$, which becomes more important when $p_t$ is time dependent.
>
> However, as described in the manuscript, we also found that the greediest strategy does not necessarily yield the best results. Therefore, we introduced delayed switching to control the degree of greediness. Although we described the mechanism only in words in the manuscript, it can be expressed as maximizing $(w_t+p_{t-1}+\delta x_t)^\top x$ in Eq. (9).
>
> In the construction of herding-based algorithm, the important point is the boundedness of $w_t$, which guarantees the $O(T^{-1})$ convergence rate and distinguishes it from probabilistic sampling with an $O(T^{-1/2})$ rate.
>
> # Proof of boundedness and convergence
> Here we provide a proof of the boundedness and convergence.
> Let $w_{t,i}$, $p_{t,i}$, $x_{t,i}$ denote the $i$th component of $w_t$, $p_t$, $x_t$, respectively.
> We first observe that the total weight $\sum_i w_{t,i}$ is conserved. Indeed, $\sum_i w_{t-1,i} = \sum_i (w_{t,i} + p_{t-1,i} - x_{t-1,i}) = \sum_i w_{t,i} + 1 - 1 = \sum_i w_{t,i}$.
> Let $\mu = \sum_i w_{t,i} / K$ be the average weight. The initial weight satisfies $w_{T,i} \ge \mu - 1$ because $w_{T,i}$ is chosen from $[0,1]$. Now assume that $w_{t,i} \ge \mu - 1$ holds for some $t$.  If $x_{t-1,i}=0$, then $w_{t-1,i} = w_{t,i} + p_{t-1,i} - x_{t-1,i} \ge w_{t,i} \ge \mu - 1$. If $x_{t-1,i}=1$, then $w_{t,i} + p_{t-1,i}$ attains the maximum value, which is at least the average $\mu$. Hence, $w_{t-1,i} = w_{t,i} + p_{t-1,i} - x_{t-1,i} \ge \mu - 1$. Thus, in both cases, we obtain $w_{t-1,i} \ge \mu - 1$, and by induction the inequality holds for all $t$. This establishes a uniform lower bound for each component of $w_t$. Since a lower bound exists and the total weight $\sum_i w_{t,i}$ is conserved, every component of $w_t$ must remain bounded from above as well, and therefore $w_t$ is bounded.
> The same argument applies to the delayed-switching version, in which case the lower bound becomes $\mu - 1 - \delta$.
>
> The boundedness of $w_t$ guarantees the $O(T^{-1})$ convergence, as $\frac{1}{T}\sum_{\tau=0}^{T-1}(p_\tau - x_\tau) = \frac{1}{T}\sum_{\tau=0}^{T-1}(w_\tau - w_{\tau+1}) = \frac{1}{T}(w_0 - w_T) \to 0$ ($T\to\infty$).

---

> > ### Author Response · Authors · 2025-11-24
> > **Improvements in reproducing the UDLM model on the LM1B dataset**
> >
> > We would also like to report our recent progress in reproducing the UDLM model on the LM1B dataset (Schiff et al., ICLR 2025). After the submission, we found a description of the training setting in the OpenReview discussion, which allowed us to partially reproduce the model with performance exceeding that of the HuggingFace model and becoming closer to the reported values. Importantly, our proposed algorithm itself remains unchanged, and our main claim that derandomization leads to improved PPL continues to hold. However, the improved reproduction now allows us to demonstrate the advantage over the baseline more clearly.
> >
> > The updated results are summarized in the new Table 1 below. Although our reproduction of the UDLM model improved, it does not yet reach the reported values. Nevertheless, with $T=128$ steps, the herding-based algorithm achieves a lower PPL than not only the original UDLM and MDLM but also the autoregressive (AR) model. Increasing the number of inference steps to $T=1024$ yields even further improvements, resulting in a PPL value (46.71) that substantially surpasses the AR model (67.46). We believe these promising results more clearly highlight the effectiveness of deterministic denoising.
> >
> > # Updated Table 1: Evaluation of LM1B text samples
> >
> > |                     | T=1024       |             | T=128        |             |
> > | ------------------- | ------------ | ----------- | ------------ | ----------- |
> > | Method              | Gen. PPL (↓) | Entropy (↑) | Gen. PPL (↓) | Entropy (↑) |
> > | AR                  | -            | -           | 67.46        | -           |
> > | MDLM                | 116.80       | -           | 120.93       | -           |
> > | UDLM                | 78.22        | -           | 79.87        | -           |
> > | UDLM (reproduced)   | 98.75        | 6.83        | 98.33        | 6.81        |
> > | UDLM herding (ours) | **46.71**    | 6.30        | **60.39**    | 6.34        |

---

> > > ### Author Response · Authors · 2025-11-24
> > > **Random samples from stochastic denoising**
> > >
> > > Below we show the first eight outputs from the UDLM model trained on the LM1B dataset using stochastic denoising.
> > > # Stochastic sampling: UDLM (reproduced)
> > > "[CLS] lukewarm to crash the world's $ 3 billion ( £1. 4 billion ) party. [CLS] it has helped to quench anger of the regular customers who long ago bought exclusive movies on the web or through itunes. [CLS] nokia has several partnerships with manufacturers to prevent the mobile phone markets from dying and plans to market several models to only the end sellers by 2010. [CLS] \" i am addicted to pain relief. [CLS] do - - vote for app print / audio. [CLS] john s. senior of the university of washington center, seattle, predicted that the previous projection of 2. 01 million jobs this year were created in 2006. [CLS]"
> > >
> > > "[CLS] suffering or the public interest, \" the minister told reporters. [CLS] in a rare move in mid - eastern virginia, stevens edged him out of warm washington politics. [CLS] between celebrating with a journey to boot the pumas have become victims of a new year of joy. [CLS] san diego fell to 6 - 0 in 25 of the 22, 000 seats at azteca stadium. [CLS] by the time they had arrived at 3. 9am a riot ensued to which no one would respond. [CLS] scandinavian investors cannot replace workers after their boards lose more than 6, 500 heads every week to other large single ventures, light of day, and today [CLS]"
> > >
> > > "[CLS]iawski, i koch ( a barquierja ), steven, ( espanola aurth ), gilbert, juan jean - baptiste ( fyzi kosian ), dujellschhofer, salim he - nazarzi, korbowu, gousset i yves heymans, pierre glover, isaiah, givat, beauxmele, plygielewski, racec, gedizzola, camoranesi, m bijzyrafors, b czemolo. [CLS] even so, rain began in the morning, left the field and put a [CLS]"
> > >
> > > "[CLS] \" it was inevitably here. [CLS] dr. h. richard hurley of the university of texas injected alexander with a tiny drug histoxipturably k to repair stuttering muscles ahead of his procedure. [CLS] face the press, so why's more? [CLS] as an outsider, oussi is still led by his armed yet steady partner. [CLS] cano drilled a solo homer in the 11th, striking the wrist of rodriguez to juice abdullah's duel with adam jones, the other in season 3 debut. [CLS] meteorologist rossio said as they tested the lake floor, temperatures found themselves hot, reaching sixc. [CLS] [CLS]"
> > >
> > > "[CLS]. [CLS] geithner also exaggerates her comments on leading the united states out of a financial crisis that has had a huge negative impact on the nation's economy, and is on the currency's creation as well as on other nations that have given power to their governments. [CLS] but their response demonstrated weakly that others might preach about direness, and some looked appropriate. [CLS] boiona de gomez of the left ( left ) did the work until her son lourdes, and son hilda of salvador, both grew up. [CLS] avery has had his tools running through 16 - inning bats and will have 385 at - bats [CLS]"
> > >
> > > "[CLS] the department of justice ; any of the actions needed may be called for the suspension of operations of any portion of the above guarana series due ( notes terminate or revoke the note ) or be required to transport the vehicle in need ; public web site resources available at website. usw. gov / aftol / increasest and ldcpp for specific. [CLS] how will his dog mane affect people? [CLS] nasa engineers said the name has been up since the 2004, considering an expected number of launches last year from the 2001. [CLS] it is also the way the latest compilation is a mistake. [CLS] christakis, [CLS]"
> > >
> > > "[CLS] brownback partners are the final participants in a half - day meeting. [CLS] unlike avid house - hunters who are more inclined to take videos of quality gadgets like web mani on toys. com, mr o'brien says regular updates on controversial tools or full encompasses of predators and a multitude of natural disasters dog other online markets. [CLS] dawn brancheau had explicitly cycled into the river seine with the remains of her dead husband. [CLS] i wonder why we do not run older companies rootcare problems. [CLS] the report showed americans are still split over u. s. economy's slow pace, deflation levels and unemployment. [CLS] a [CLS]"
> > >
> > > "[CLS] the discussion. - consolidatedtable conclusions both as a historical and comprehensive transaction. sales, revenue and revenues for moving forward in one empty but titled year. been in filings on form s - 8b filed by the sec with the securities and exchange. [CLS] they are normally spent during trials at the court itself. [CLS] djokovic quickly began his resurgence, winning an emphatic ninth game as federer hauled down three break points. [CLS] organised crime, the always, streaked this true. [CLS] \" all cities will benefit, \" said niamh myung norungon of the university of california, san diego and colleagues at the [CLS]"

---

> > > > ### Author Response · Authors · 2025-11-24
> > > > **Random samples from deterministic denoising**
> > > >
> > > > Below we show the first eight outputs from the same UDLM model trained on the LM1B dataset using deterministic denoising.
> > > > # Deterministic sampling: UDLM herding (ours)
> > > > "[CLS] renault - nissan alliance, which has been with the white house. [CLS] so check the scrape to tell you what you can... you can. [CLS] the first one you saw, that was called \" the doctrine of the doctrine of the faith, what right? \" [CLS] who is in damage control? [CLS] \" no one wants to shoot at a building next to the supreme court, \" he said. [CLS] this has been the only long - running issue - that we turned up for the t20 world championship instead of ashes series and i'm sure that it's going to be the number two decision. [CLS] as the [CLS]"
> > > >
> > > > "[CLS] get there, i won't pass it up, but i would prefer to be in the country in 2009, \" he said. [CLS] so what does this new policy do? [CLS] not even bad, but obama, who had just phoned bush to say it was real, had no need lose himself in a politician's badge of honor. [CLS] the young man's years of abuse started at an age in 1979 when he started praying at the same mosque in birmingham. [CLS] a force spokeswoman said : \" while levels of crime at the premises have fallen, it is our responsibility to make sure we can continue these [CLS]"
> > > >
> > > > "[CLS], she said she and her husband will spend time together after the game in cleveland. [CLS] \" we are in progress with confidence, with the strength of our 1752 portfolio and the ability to meet our obligations to the market, \" he said. [CLS] all five of the banking groups have promised support for a scheme worth more than £50 billion, 69 % of which was borrowed before the tax change in april. [CLS] his father told the priest that the boy would come out of retirement but would only return home at the end of his life. [CLS] he was also involved in the crash. [CLS] if you happen to wait for the day, [CLS]"
> > > >
> > > > "[CLS] is any way to stop, speak to me. [CLS] it said that it was channeling shareholders through a \" period of difficult trading \" that had been far short of expectations. [CLS] \" access becomes an issue for the children every time we get out of the school, \" the spokesman said. [CLS] \" he really is never going to settle down. [CLS] she was the proud parent of three of her children who said it was a nightmare. [CLS] davis said bush's words helped ease her train ride. [CLS] samples of the water, thought to have been drained due to sewage, would be available in the area over three weeks. [CLS] [CLS]"
> > > >
> > > > "[CLS] near - constant tension in darfur, one of the world's poorest, about 60 miles east of khartoum. [CLS] taken from the gses'perspective, \" the urban unemployment rate has dropped 30 percent in the past year, not by more than 7 percent, but only by 5 percent, \" the group wrote in the original report, which favored health, safety and financial standards. [CLS] \" i'd like you to, \" he said. [CLS] \" we have the power, one agency at a time... respond to a new law, \" he assured congress. [CLS] o. k. the axle [CLS]"
> > > >
> > > > "[CLS] battle of her own in the us election for the nomination. [CLS] also, officials say 911 callers confirm that the caller is related to a suspect in the vicinity. [CLS] there is no immediate prospect that fiat motors, a large u. s. - owned subsidiary that competes with the italian automaker fiat, will take over operations of its michigan plant, gm said. [CLS] former lover michelle roberts, 31, called the matter a long - distance affair. [CLS] and since foreign troops came to their assistance early in 2007, the afghan authorities have faced a growing flood of taliban military forces. [CLS] it offers a lot of features, \" said [CLS]"
> > > >
> > > > "[CLS] [CLS] and she says : i don't think she's very lucky but today she's feeling better. [CLS] prices are still falling. [CLS] this would be microsoft's biggest buy - out exercise ever in yahoo's long history, but executives say microsoft could have time to turn it around in just over a year. [CLS] just when the war was under way, u. s. forces were in iraq for the entire year and spent nearly four times the u. s. budget for operations in their adopted country in 2003, 2004 and october 2007. [CLS] it also includes the vast majority of the big east's major [CLS]"
> > > >
> > > > "[CLS] today's economy, new schools and more flexibility, \" duncan said. [CLS] and, yes, we were confused. [CLS] the next chapter represents an important move toward the right. [CLS] and as it stands today, i hesitate to say this because the administration still has a specific intent to put the obama administration's unemployment rate at 7 million to 9. 7 million. [CLS] comment : some times he was able to take an on - line gender issue for political gain... maybe he was just asking why. [CLS] you know what, we must change if we all want to understand this. [CLS] \" there is no time for [CLS]"

---

### Author Response · Authors · 2025-12-03
**Summary of the changes in the revised manuscript**

We have revised our manuscript with sincere respect for the efforts of the reviewers and ACs, both former and current. While all reviewers acknowledged the novelty and originality of our approach, which introduces the herding algorithm into discrete diffusion models, they also raised concerns about the clarity of our exposition, specifically: the discussion of its relation to prior work (ezsi), the applicability of our approach (uEJ8), the evaluation of generated samples (uEJ8 & hA6i), and the explanation of the proposed algorithm (uEJ8, hA6i & B8Lv). The changes are summarized as follows.
- We describe the proposed time-dependent herding algorithm in more detail and provide a mathematical proof of boundedness and convergence.
- We have improved our reproduction of the UDLM model trained on the LM1B dataset and now present substantially better PPL values in Table 1, which outperform the AR model.
- As for sample quality, we include the MAUVE score in Table 1, which indicates that our algorithm affects distributional metrics only minimally. We also provide random text samples generated by the UDLM model trained on the LM1B dataset.
- We have improved the discussion in the context of the prior literature on discrete diffusion models, including the points raised by Reviewers ezsi and uEJ8, with additional citations covering most of the valuable references suggested by Reviewer ezsi.

It is unfortunate that we are unable to receive further comments from the reviewers. However, we believe that all of the concerns they raised have now been addressed reasonably.

---

### Meta-Review · Area_Chair_HdDA · 2026-01-07

**Summary:**

The authors propose a deterministic denoising algorithm for discrete diffusion models, which serves as a discrete counterpart to DDIM. Instead of relying on stochastic sampling methods, the approach draws on the herding algorithm and concepts from chaotic dynamical systems to construct a deterministic reverse diffusion process. The method does not require training, works with any discrete diffusion model using uniform noising, and uses randomness only in the initial noise state. Performance is demonstrated on text and image generation.

The initial recommendations for this manuscript were 1 marginally below the acceptance threshold (ezsi), and 3 rejects (uEJ8, hA6i, B8Lv). Reviewers noted that the manuscript has clear motivation and provides an interesting angle. Additionally, reviewers also noted that the herding algorithm has not been applied in the this context before, which constitutes novelty.

Nonetheless, the reviewers raised a number of questions and concerns:
(1) lack of discussion of theoretical foundations and analysis of the herding algorithm in the context of discrete diffusion models (ezsi);
(2) missing baselines (ezsi); (3) generalization of the methodology to LLaDA and whether it relates to standard algorithms like parallel/speculative decoding in LLMs (ezsi); (4) how the dynamics relate to the inference-time scaling framework for diffusion models (ezsi); (5) limited scope of applicability in that the proposed method is only effective for uniform discrete diffusion models (uEJ8); (6) missing motivation of studying deterministic samplers (uEJ8); (7) theoretical guarantees on the convergence of Eqs. 7, 8 (uEJ8); (8) clarifications on Eqs. 7, 8 and Eqs. 9, 10 (uEJ8, B8Lv); (9) lack of qualitative results, pseudo code, related section on herding algorithms (uEJ8, ezsi, hA6i, B8Lv); (10) lack of explanation for technical concepts despite ample space in the main paper, e.g., delayed switching, (hA6i, B8Lv); (11) generative perplexity alone cannot be regarded as a good metric...using the MAUVE score instead (hA6i).

The authors were able to address most of the questions and concerns raised. However, a critical point was shared amongst reviewers in that the manuscript currently is lacking discussion on related works, clarification of concepts, pseudo code and discussion of algorithm, qualitative examples, etc. While the AC believes that the authors have sufficiently addressed the technical questions, incorporation of the materials as well as content not included in the rebuttal requires a major revision. The AC agree with the reviewers that the presentation of manuscript is currently not suitable for publication. However, the results and additional discussion presented in the rebuttal will strengthen the manuscript. The AC recommends the authors to incorporate the feedback and materials into the next revision and resubmit to another top venue.

**Reviewer Concerns:**

The authors posted a rebuttal to address the following points:

(1) lack of discussion of theoretical foundations and analysis of the herding algorithm in the context of discrete diffusion models (ezsi):

The authors have incorporated related works and additional discussion on herding algorithm and discrete diffusion models.

(2) missing baselines (ezsi):

The authors included DNDM as an additional experiment. This point is addressed.

(3) generalization of the methodology to LLaDA and whether it relates to standard algorithms like parallel/speculative decoding in LLMs (ezsi):

The authors responded that there needs to be a further generalization of the method to masked diffusion. This question has been addressed.

(4) how the dynamics relate to the inference-time scaling framework for diffusion models (ezsi):

The authors responded that there is the time dependent scheme does induce a flow of probability. mass over sample space and could be related to gradient flow. The question has been addressed.

(5) limited scope of applicability in that the proposed method is only effective for uniform discrete diffusion models (uEJ8):

The authors responded that while masked diffusion is the current trend, their work is positioned for a broader scope and the possibility of future developments and contribution towards the trend. The AC agrees with the authors. The point has been addressed.

(6) missing motivation of studying deterministic samplers (uEJ8):

The authors noted the convergence rate of deterministic herding-based sampler as the primary motivation and provided a proof of convergence in the comment. This point has been addressed.

(7) theoretical guarantees on the convergence of Eqs. 7, 8 (uEJ8):

The authors provided a proof in the comment. This points has been addressed.

(8) clarifications on Eqs. 7, 8 and Eqs. 9, 10 (uEJ8, B8Lv):

The authors provided clarifications on Eqs. 7, 8 and Eqs. 9, 10 in the comment. This point has been addressed.

(9) lack of qualitative results, pseudo code, related section on herding algorithms (uEJ8, ezsi, hA6i, B8Lv):

The authors mentioned that they will provide such in the next revision of the manuscript. The AC notes that the revised manuscript does not have qualitative examples. Additionally, addressing all of the comments from the reviewers may require a major revision of the manuscript.

(10) lack of explanation for technical concepts despite ample space in the main paper, e.g., delayed switching, (hA6i, B8Lv):

The authors provided an explanation of time-dependent version and delayed switching in the comment. This point has been addressed.

(11) generative perplexity alone cannot be regarded as a good metric...using the MAUVE score instead (hA6i):

The authors included MAUVE as a metric in Table 1 and provided random text samples generated by the UDLM model trained on the LM1B dataset. This point has been addressed.

**Reviewer Scores:**

The AC read the reviews and rebuttal. Based on the responses, the AC believes that all reviewers may raise their scores as most of the technical questions have been addressed. However, the AC believes that uEJ8, hA6i, and B8Lv will remain negative in their recommendation as the manuscript will require a major revision.

---

### Decision · Program_Chairs · 2026-01-26

Reject